# In Vitro Production and Identification of Angiotensin Converting Enzyme (ACE) Inhibitory Peptides Derived from Distilled Spent Grain Prolamin Isolate

**DOI:** 10.3390/foods8090390

**Published:** 2019-09-04

**Authors:** Dong Wei, Wenlai Fan, Yan Xu

**Affiliations:** Key Laboratory of Industrial Biotechnology of Ministry of Education, Laboratory of Brewing Microbiology and Applied Enzymology, School of Biotechnology, Jiangnan University, Wuxi 214000, Jiangsu, China

**Keywords:** distilled spent grain (DSG), baijiu (Chinese liquor), ACE inhibitory peptides, Prolamin isolate hydrolysates, RP-HPLC, AVQ, YPQ

## Abstract

Distilled spent grain (DSG), the biggest by-product of the Chinese liquor industry, is rich in protein (167.8 g/kg DSG dry weight (DW)). Accounting for 60% of the total protein, prolamins are isolated from dried DSG (DDSG). In this study, angiotensin-converting enzyme (ACE) inhibitory peptides were screened from the prolamin hydrolysates of DDSG using two independent active-directed separations, ultrafiltration and reversed phase high performance liquid chromatography (RP-HPLC) coupled with ACE inhibitory activity evaluation. Six novel ACE inhibitory peptides, AVQ, YPQ, NQL, AYLQ, VLPVLS, and VLPSLN, were successfully identified and quantified from the active RP-HPLC fractions. AVQ and YPQ exhibited the highest activity, having the concentration inducing 50% inhibition (IC_50_) values for ACE of 181.0 and 220.0 μM, respectively. It was observed that VLPVLS was the most abundant peptide (16.96 mg/g DW) in prolamins. The results indicated that prolamin hydrolysates from DDSG could be served as a source of ACE inhibitory peptides.

## 1. Introduction

Hypertension is one of the well-defined risk factors of cardiovascular disease, with about 30% adults worldwide suffer from high blood pressure [1,2]. Angiotensin converting enzyme (ACE, EC 3.4.15.1), a zinc-containing dipeptide carboxypeptidase, is a key rate-limiting enzyme for the regulation of blood pressure through the renin-angiotensin system (RAS) and kinin–kallikrein system (KKS) in living organisms. As part of RAS, ACE hydrolyzes an inactive angiotensin I to vasoconstrictor angiotensin II while ACE inactivates the hypotensive bradykinin with strong vasodilation in the KKS [3,4]. Hence, ACE inhibitors were expected to inhibit the formation of the vasoconstrictor angiotensin II and enhance bradykinin vasodilation (hypotension) properties [5].

Food-derived ACE inhibitors such as peptides with high ACE inhibitory activity are believed safer than drugs with related side effects, such as cough and angioedema [6]. In order to a continuous dietary intake, ACE inhibitory peptides derived from food proteins are expected to be isolated. The most strategy to induce the release of antihypertensive peptides in vitro is enzymatic digestion (hydrolysis). Since the discovery of the first ACE inhibitory peptide in caseins [7], a variety of bioactive peptides with ACE inhibition have been found in enzymatic hydrolysates of proteins from different sources, such as caseins and whey proteins [8,9], algae proteins [10], rice bran protein [11], brewers’ spent grain [12], and soybean [13]. The development of new functional foods bonded with ACE inhibitory peptides from natural protein sources may be an important part of the prevention against hypertension. Meanwhile, to date, in by-product of the alcoholic industry, only studies on ACE inhibitory peptides in brewers’ spent grain [12] and sake lees [14,15] have been reported.

Distilled spent grain (DSG), a major by-product of the baijiu (Chinese liquor) industry, reaches annual output of appropriately 30 million tons. The main materials for making baijiu are sorghum (accounted for 80% w/w) and wheat (20% w/w). DSG was mainly used for producing feed and biogas [16], now. With the exceptions of carbohydrates (5.71–11.34%), crude fiber (10.05–10.20%), crude lipids (1.31–3.24%), and protein (5.0–13.8%). Sorghums for brewing are the predominant raw materials, and whose prolamins and kafirin account for 50–60% of the total protein [17]. Functional properties of plant proteins, including ACE inhibitory activity, would be improved by hydrolysis with addition of endoprotease [18]. It has been reported that lots of water-soluble peptides with physiological activity were identified from DSG by Wei et al. [19]. However, the potential for the generation of ACE inhibitory peptides from dried DSG (DDSG) protein has not been investigated so far.

To illustrate the potential of a DDSG prolamin isolate (DDSG-PI) as the raw material for generation of ACE inhibitory peptides, in this study, the potential ACE inhibitory effects of prolamin hydrolysates of DDSG-PI were evaluated, and ACE inhibitory peptides were purified by ultrafiltration and semi-preparative reversed phase high performance liquid chromatography (RP–HPLC). Ultra-performance liquid chromatography (UPLC) coupled with quadrupole time-of-flight mass spectrometry (Q–TOF–MS) was employed for peptide identification, and the quantification for target peptides was established by triple-quadrupole mass spectrometer (QQQ–MS) with an electrospray ionization source (ESI). The utilization of DDSG for generating ACE inhibitory peptides could be an efficient attempt to gain economic and environmental advantage.

## 2. Materials and Methods

### 2.1. Materials

DSG samples used in this study were kindly provided by Golden Seed Distillery Co., Ltd. (Fuyang, Anhui, China) and stored at −20 °C until use.

Alcalase 2.4 L (EC 3.4.21.62, 2.4 Anson units/g) was purchased from Sigma–Aldrich Chemical Co. (Shanghai, China). Neutrase 0.8 L (EC 3.4.24.28, 0.8 Anson units/g) and Flavourzyme 500 MG (EC 3.4.15.1, 1500 leucine aminopeptidase units/g) was purchased from Novozymes (Bagsvaerd, Denmark). Acetonitrile of HPLC grade was obtained from Adamas Reagent, Ltd. (Shanghai, China). DL-Dithiothreitol (DTT, 99% purity) and *o*-phthaldialdehyde (OPA, 97% purity) was purchased from Aladdin Bio-Chem Technology Co., Ltd. (Shanghai, China). All peptide standards were synthesized in our lab. All other chemicals were of HPLC or analytical grade.

### 2.2. Extraction of Prolamins from DDSG

The DSG samples were removed from −20 °C, thawed at room temperature and oven-dried at 70 °C for 24 h to a constant weight. The DDSG was obtained by removing the rice husk. DDSG was fine-ground to flour using a laboratory mill with a 0.25 mm mesh screen (Sinopharm Chemical Reagent Co., Ltd., Hefei, Anhui, China).

DDSG-PI was isolated using the NaOH-ethanol method as previously described [20] with some modifications. The DDSG flour (150 g) was mixed with 1.05 L 70% (by volume) ethanol–water solution, 0.35% (w/v) NaOH, and 0.5% (w/v) sodium metabisulfite solutions. This solution was stirred 1 h at 70 °C in water bath, and centrifuged at 3500 *g* for 10 min. Then the ethanol in the supernatant was diluted to 40% (by volume) and retained at −20 °C overnight. The suspension was centrifuged at 3500 *g* for 10 min, and the sediment was rinsed three times using deionized water by centrifuging 5000 *g* for 10 min. Then, the sediment was lyophilized (Labconco Corp., Kansas City, MO, USA). The DDSG-PI was obtained by mixing lyophilized DDSG with 5-fold weight of *n*-hexane by shaking at 70 rpm for 10 min. After centrifugation (5000 *g* for 10 min), the sediment was collected. Defatting procedure was repeated three times. The sediment was dried at 50 °C for 12 h, and then retained at −20 °C for further analysis.

### 2.3. Protein Content Analysis

Protein content of DDSG-PI was measured using macro-Kjeldahl procedure as previously described [21]. A value of 5.8 was set as protein conversion a factor.

### 2.4. Amino Acid Composition Analysis

The amino acid composition of DDSG-PI was determined using an Agilent1100 HPLC system (Agilent Technologies, Inc., Santa Clara, CA, USA) with UV detection. DDSG-PI was hydrolyzed by 6 N HCl for 22 h at 120 °C followed by filling with nitrogen for 3 min. Neutralized with 10 M NaOH, 1 mL of sample was centrifuged at 15,000 *g* for 30 min prior to OPA pre-column derivatization. Derivatized sample was injected into a Hypersil ODS column (250 × 4 mm, 5 μm, Agilent Technologies, Inc., Santa Clara, CA, USA). The mobile phase A was consisted of 27.6 mM sodium acetate: trimethylamine: tetrahydrofuran at a ratio of 500:0.11:2.5 (v/v/v, pH 7.2), while mobile phase B was 80.9 mM sodium acetate:acetonitrile:methanol, at a ratio of 1:2:2 (v/v/v, pH 7.2). Then, the derivatized amino acids were separated at a flow rate of 1 mL/min, using the followed linear gradient: 0 min, 8% B; 0–17 min, 50% B; 17.0–20.1 min, 100% B; 20.1–24.0 min, 0% B. The absorbance was monitored at 338 nm and 262 nm. A standard solution of 17 amino acids was used as external standard.

### 2.5. Enzymatic Hydrolysis of DDSG-PI

The DDSG-PI hydrolysates were prepared in sealed reaction vessel under controlled conditions using the OPA method [22]. The defatted DDSG-PI was re-suspended in 0.01 M phosphate buffer at a ratio of 1:20 (w/v) and allowed to hydrate using an overhead stirrer. The defatted DDSG-PI was separately hydrolyzed for 10 h by three types of proteases according to their respective optimum conditions using an enzyme/substrate ratio of 1:10 (w/w, based on the prolamin content of defatted DDSG-PI) (Alcalase 2.4 L, pH 8.0, 55 °C; Neutrase 0.8 L, pH 7.0, 45 °C; Flavourzyme 500 MG, pH 8.0, 55 °C). At different time-points, 1 mL of hydrolysate samples were withdrawn for further analysis. To cease the reaction, this hydrolyzed mixture was heated at 90 °C for 15 min to inactivate enzymes. The hydrolysate was then centrifuged at 5000 *g* for 15 min to separate the supernatant and insoluble residues. Finally, the supernatant was rapidly cooled at room temperature and stored at −20 °C for further use.

### 2.6. Determination of the Degree of Hydrolysis (DH)

DH, defined as the ratio of peptide bonds hydrolyzed (h) to total peptide bonds in protein, was determined by the OPA method [23] with some modifications. The OPA reagent was prepared as follows: 7.620 g sodium tetrahydroborate and 200 mg sodium dodecyl sulfate (SDS) were dissolved in 150 mL deionized water as solution 1. The solution 1 should be completely dissolved before adding other reagents. 160 mg of OPA was dissolved in 4 mL ethanol as solution 2. The solution 2 was then quantitatively mixed to the solution 1 by rinsing with deionized water, which was defined as solution 3. Then, 176 mg of DTT was added to the solution 3 followed by adding with deionized water to made up to 200 mL.

Before the measurement, 1 mL of hydrolysate sample was necessarily diluted to 100 mL with deionized water. In order to measure the DH, 400 μL of the diluted sample was mixed with 3 mL of OPA reagent in a cuvette. The mixture was incubated in the dark for 2 min. Then, the absorbance was recorded at 340 nm using a UV spectrophotometer (A380, AOYI instruments co., Ltd., Shanghai, China) and the deionized water was set as the control. In order to convert the absorbance readings into DH, a serine standard curve with appropriate concentrations was established, with *R*^2^ 0.99, using the equation previously described [24].

### 2.7. Determination of ACE Inhibitory Activity

The ACE inhibitory activity was determined following the method as previously described, using the colorimetric detection system (ACE kit-WST, Dojindo Inc., Kumamoto, Japan). The assay procedure was consistent with the technical manual provided by the assay kit. Briefly, 20 µL of sample solution was added to a sample well, and 20 µL of deionized water was added to blank 1 and blank 2 wells. 20 µL of substrate buffer was added to each well. 20 µL of deionized water was added to blank 2 well, while 20 µL of enzyme working solution was added to each sample and blank 1 wells. All wells containing above solutions were incubated at 37 °C for 1 h. Afterwards, 200 µL of the indicator working solution was added to each well with further incubation for 10 min. Then, the absorbance was recorded at 450 nm using a microplate reader (Cytation 3, BioTek Instruments, Inc., Winooski, VT, USA). The ACE inhibitory activity was calculated using the equation
(1)$$\mathrm{ACE}\;\mathrm{inhibitory}\;\mathrm{activity}\;\left(\mathrm{inhibition}\;\mathrm{rate}\;\%\right)\;=\;\frac{A_{\mathrm{blank}1}-{\;A}_{\mathrm{sample}}}{A_{\mathrm{blank}1}-{\;A}_{\mathrm{blank}2}}\;\times \;100\%$$
where A_blank1_ represented positive control (without ACE inhibition); A_blank2_ was the absorbance of the reagent blank, while A_sample_ was the absorbance of the sample. The concentration inducing 50% inhibition (IC_50_) of peptides and some samples were also determined with non-linear regression.

### 2.8. Ultrafiltration of DDSG-PI Hydrolysates

The Alcalase-generated hydrolysate (AGH) from DDSG-PI was separated by using three different cutoff cellulose ultrafiltration membranes (Millipore Corp., Billerica, MA, USA) at molecular mass cutoff values of 5, 3, and 1 kDa. The four fractions with MW >5 kDa, 3–5 kDa, 1–3 kDa, and <1 kDa were obtained by using the Labscale TFF system kit (Millipore Corp., Billerica, Massachusetts, USA) and the fixture (Millipore Corp., Billerica, MA, USA). The four fractions were lyophilized and stored at −20 °C until further analysis.

### 2.9. Semi-Preparative RP–HPLC

According to the method previously described [12], the lyophilized fraction obtained from <1 kDa permeate of AGH, was further separated by using semi-preparative RP-HPLC (Waters Corp., Milford, CT, USA). The <1 kDa fraction was dissolved in deionized water with the final concentration of 5 mg/mL. 800 μL of aliquots were injected into an XBridge BEH-C18 semi-preparative column (250 × 10 mm, 5 μm, Waters Corp., Milford, CT, USA) with a C18 guard column (Waters Corp., Milford, CT, USA). The mobile phase A was made up of Milli-Q water (Millipore, Bedford, USA), while mobile phase B was 80% (v/v) acetonitrile. During each of 25 runs, the fraction (<1 kDa permeate) was separated at a flow rate of 2 mL/min and eluted with the followed linear gradient: 0–4 min, 5% B; 4–8 min, 5–21% B; 8–45 min, 21–45% B; 45–55 min, 45–55% B; 55–56 min, 55–100% B; 56–60 min, 100% B; 60–61 min, 100–5% B; and 61–71 min, 5% B. Elution was monitored at 214 nm. Afterwards, the collected fractions were pooled into eight fractions, I–VIII (6.0–11.1 min, 11.1–17.3 min, 17.3–21.5 min, 21.5–23.5 min, 23.5–26.2 min, 26.2–32.5 min, 32.5–42.0 min, and 42.0–50.0 min). The collected fractions I–VIII were lyophilized for the measurement of ACE inhibitory activity.

### 2.10. Peptide Identification by UPLC–Q–TOF–MS/MS

The active ACE inhibitory fractions II–VIII obtained from semi-preparative RP-HPLC were dissolved in deionized water and reconstituted to 1 mg/mL. Peptide identification was carried out by Acquity UPLC system with a BEH130 C18 analytical column (150 × 2.1 mm, 1.7 μm, Waters Corp., Milford, USA), coupled to a SYNAPT Q–TOF mass spectrometer (Waters Corp., Milford, CT, USA). 1 μL of sample was separated with mobile phase A (0.1% formic acid in deionized water) and mobile phase B (acetonitrile) at a flow rate of 0.3 mL/min, with a linear gradient as follows: 0–5 min, 0% B; 5–20 min, 0–20% B; 20–22 min, 20–60% B; 22–25 min, 60–100% B; and 25–25.1 min, 100–0% B.

Positive ion mode was used in the mass spectrometer equipped with an ESI source. The precursor ions were selected from MS spectra, which were produced by full scan-survey mode ranging from 50 to 1500 Da. Then, the target ions were fragmented by collision-induced dissociation (CID, MS/MS mode). Masslynx V4.1 software (Waters Corp., Milford, CT, USA) was utilized to operate the identification process and analyze the MS and MS/MS spectra. The optimized conditions of the Q–TOF mass spectrometer are presented in Table 1. The MS/MS data were processed by de novo sequencing using a Biolynx peptide sequencer (Micromass UK Ltd., Manchester, UK) in this experiment. The input parameters required in Biolynx peptide sequencer were the peptide molecular weight (precursor ion) based on analysis of MS and MS/MS spectra, monoisotopic MW (0.5), the precursor charge (1), and fragment ion tolerance (0.3, monoisotopic), while other parameters were default values.

### 2.11. Peptide Synthesis

Peptides were synthesized by using an automated microwave peptide synthesizer (LibertyBlue, CEM Corp., Matthews, NC, USA). The synthesized peptides were purified by HPLC. The molecular masses of peptides were identified by a Q–TOF mass spectrometer. The purity of peptides was more than 98%. The synthesized peptides were stored at −20 °C for further use.

### 2.12. Quantification of Peptides with ACE Inhibitory Activity

According to the protocol described by Steffi et al. [25], the identified peptides with potential ACE inhibitory activity were quantified by Acquity UPLC system coupled with a QQQ–MS (Waters Corp., Milford, CT, USA) with an ESI source. Multiple reaction monitoring (MRM) mode was used for optimization of mass spectrometry parameters of individual peptides with a BEH C18 analytical column (100 × 2.1 mm, 1.7 μm, Waters Corp., Milford, CT, USA). Mobile phase A was formic acid (0.1% in Milli-Q water) and mobile phase B was acetonitrile (0.1% formic acid). The mobile phase gradient and flow rate were consistent with the qualitative steps described in the section “Peptide identification by UPLC–Q–TOF–MS/MS”. ESI parameters were set as follows: positive mode; desolvation temperature, 350 °C; desolvation gas flow, 800 L/h; capillary voltage, 3500 V. MRM transitions were selected by Intellistart, the plugin of Masslynx V4.1 which would generate the optimal product ion and corresponding collision energy. The synthetic peptide standards containing AVQ, NQL, YPQ, AYLQ, VLPVLS, and VLPSLN were set as external standards.

### 2.13. Statistical Analysis

All tests in this study were replicated three times. The statistical analysis and comparison were presented by one-way ANOVA using SPSS 19.0 version (SPSS Inc., Chicago, IL, USA), where significant differences were set as significant at *p* < 0.05. The results were expressed as mean ± standard deviation (SD).

## 3. Results and Discussion

### 3.1. Prolamin Content of DDSG and Its Amino Acid Composition

The results of the Kjeldahl method showed that the prolamin average content was 45.52 mg/g DDSG DW. The total amino acid composition was 940.0 mg/g DDSG-PI DW (Table 2). Obviously, Glu was the most abundant component (250.8 mg/g DDSG-PI DW), followed by Leu (154.3 mg/g DDSG-PI DW), Ala (98.50 mg/g DDSG-PI DW), and Pro (88.50 mg/g DDSG-PI DW), while the concentrations of Cys and Lys were extremely low, less than 0.50 mg/g. DDSG has a significant amount of hydrophobic amino acids. According to Cheung et al.’s description [26], high content of hydrophobic amino acids could have potential ACE inhibitory activity due to C-terminal amino acids of ACE inhibitory peptides, such as Pro, Ala, and Leu. Therefore, DDSG-PI could be a source of bioactive peptides with ACE inhibitory activity.

### 3.2. DH Analysis of DDSG-PI Hydrolysates

One of the important parameters for affecting ACE inhibitory activity was DH, which was usually calculated by OPA method [27,28]. As shown in the Figure 1, the DH of Alcalase hydrolysate was higher (*p* < 0.05) compared to other two hydrolysates during the same hydrolysis times. During 5 h of hydrolysis, Alcalase hydrolysates showed the highest DH at 15.5%, while the DH values of Flavourzyme and Neutrase hydrolysates were 4.70% and 10.1%, respectively. The DH increased with hydrolysis time while the hydrolysis of Alcalase and Flavourzyme was ceased nearly at 5 h, an indication that enzymatic hydrolysis was saturated. This may be due to the high concentration of hydrolysate which showed feedback suppression on the activity of the enzyme, or the concentration of available peptide bonds decreased as the hydrolysis proceeded [29]. The results showed that Alcalase was more effective compared to the other two enzymes.

### 3.3. ACE Inhibitory Activity with Hydrolysis Time

The screening of DDSG-PI hydrolysates for ACE inhibitory capacity was crucial in this study. The enzymatic hydrolysis was critical for the release of ACE inhibitory peptides from DDSG-PI (Figure 2). Among three enzymatic hydrolysates, Alcalase hydrolysates showed the highest ACE inhibitory activity value of 79.05% (Figure 2C) at 3 h of hydrolysis time, followed by hydrolysates (67.58%) (Figure 2B) generated by Neutrase at 4 h and Flavourzyme hydrolysates (42.84%) (Figure 2A) at 4 h.

During 5 h hydrolysis, the ACE inhibition values for different proteases decreased slightly after a certain time point, indicating that the released ACE inhibitory peptides could be further hydrolyzed as a substrates. Moreover, it could be concluded that there was no significant correlation between DH and ACE inhibition. That was consistent with what Mao et al. [30] reported.

### 3.4. ACE Inhibitory Activity of Ultrafiltration Fractions

Ultrafiltration of the AGH was performed in order to find the fraction with higher ACE inhibitory activity. The AGH obtained from DDSG-PI was fractionated by ultrafiltration with 5 kDa, 3 kDa, and 1 kDa membrane to get four fractions with >5 kDa, 3–5 kDa, 1–3 kDa, and <1 kDa. Each fraction generated from ultrafiltration was freeze-dried and dissolved in deionized water to a final concentration of 1 mg/mL, and the ACE inhibitory activity of each fraction was determined by a gradient dilution (five-fold) to obtain IC_50_ values.

The fractions of ultrafiltration with molecular weights >5 kDa, 3–5 kDa, and 1–3 kDa showed low ACE inhibitory activity with IC_50_ values of 24.58, 10.41, and 26.74 μg/mL, respectively. However, the fraction with <1 kDa exhibited the most potent ACE inhibitory activity (IC_50_ value of 6.750 μg/mL) and the highest yield of 84.18% (Table 3).

It was obviously observed that the ACE inhibitory activity of ultrafiltration hydrolysate herein was not directly correlated with molecular weight distribution range (Table 3), which was consistent with the findings reported previously [12,31]. Connolly et al. [12] revealed that fractions of AGH from brewers’ spent grain protein with molecular weights 1–5 kDa exhibited highest ACE inhibition, followed by 5–10 kDa and <1 kDa.

Due to the most active ACE inhibitory activity and highest yield, the <1 kDa permeate was selected to purify by further fractionation.

### 3.5. Semi-Preparative RP-HPLC Fractionation of AGH

RP-HPLC can reach the separation effect of peptides in terms of the hydrophobicity of compounds [32]. The <1 kDa permeate from AGH, which showed highest ACE inhibitory capacity, was fractionated by semi-preparative RP-HPLC and the elution profile was presented in Figure 3. The pooled fractions II–VIII were found to possess high ACE inhibitory property (73.29–88.99% ACE inhibition), of which fraction VI exhibited the highest ACE inhibition (88.99%), at approximately 28% acetonitrile. Fraction I showed the lowest ACE inhibition among all pooled fractions.

Most of peptides with potential ACE inhibitory activity herein were collected from fractions eluted at approximately 18–40% acetonitrile. Moreover, the ACE inhibition increased with the increasing concentration of acetonitrile but until about 28% acetonitrile, after which ACE inhibition began to decline (Figure 4). The percentage of acetonitrile could indirectly characterize the hydrophobicity of the eluent. The hydrophilic molecules eluted at early fraction, while the later eluted fraction was composed of more non-polar or hydrophobic molecules. It had been reported that fractions, showing the high ACE inhibitory activity, generally resulted in higher retention in RP-HPLC runs, which refers to peptides with high amounts of hydrophobic amino acid residues [33,34]. Thus, pooled fractions II–VIII with high ACE inhibitory activity were selected for further peptide sequencing by UPLC–Q–TOF–MS/MS.

### 3.6. Identification of Peptides and Their ACE Inhibitory Activity

The active ACE inhibitory fractions II–VIII were analyzed by UPLC–Q–TOF–MS/MS and the peptide characterization was based on their MS and MS/MS spectra, which could provide information of accurate precursor and product ions. Most peptides were usually protonated under ESI conditions. In the low collision energy (<200 eV) environment, the chemical bond cleavages of peptides occurred primarily at the amide bonds [35]. Therefore, the b and y ions were the main fragment ions.

Six novel ACE inhibitory peptides (AVQ, NQL, YPQ, AYLQ, VLPVLS, and VLPSLN) could be identified by UPLC–Q–TOF–MS/MS in this study. Among these, AYLQ was selected as an example to illustrate the identification process of peptides. The mass spectra with different collision energy (6 eV and 20 eV) were analyzed herein to identify the precursor ion. It was observed that at the same retention time, the abundance of *m/z* 494.2571 at a collision energy of 20 eV (Figure 5A) was relatively low compared to the collision energy of 6 eV (Figure 5B), and the abundance other fragment ions raised. Moreover, the ion *m/z* 987.5105 (Figure 5A) was proved to be the [2M + H] + ion. Therefore, *m/z* 494.2571 was regarded as the precursor ion.

The MS/MS spectrum of *m/z* 494.2571 is shown in Figure 6 (only b and y fragment ions are shown). Fragment ion *m/z* 147.0756 was proven to be a y1 ion and also represented the [Gln + H] +, while *m/z* 348.1922 was regarded as a b3 ion and considered to be a [M − y1 + H] + ion. The b2 ion (*m/z* 235.1100) was the fragment [Ala−Tyr−H_2_O + H] +, so the y2 ion was *m/z* 260.1613. The fragment ion *m/z* 423.2319 was proved to be a y3 ion, while the b1 ion has not been observed. In fact, b1 ions were rarely observed [36]. Based on the above analysis, fragments between y1 and y2 ions were regarded as leucine residue, and tyrosine residue was between y2 and y3 ions. By comparing the retention time and fragment ions of the standard, thus, the peptide sequence was determined as AYLQ.

The results showed that 22 peptide sequences were identified in these fractions, of which six novel peptide sequences were determined, based on the literature search, AHTPDB-UWM (http://www.crdd.osdd.net/raghava/ahtpdb/) [37], and BIOPEP-UWM (http://www.uwm.edu.pl/biochemia/index.php/en/biopep) [38].

According to the results of the IC_50_ for ACE (Table 4), it was observed that AVQ exhibited the highest ACE inhibitory capacity with its IC_50_ values of 181.0 μM, followed by YPQ (220.0 μM), AYLQ (228.6 μM), VLPVLS (248.1 μM), NQL (704.6 μM), and VLPSLN (1246 μM).

Although the relationship between the structure of the ACE inhibitory peptides and the ACE inhibition properties has not been thoroughly studied, some common features of the ACE inhibitory peptides have been summarized. Peptides with low molecular weights and short sequences (2–12 amino acid residues), or hydrophobic amino acids at the C-terminal position could enhance the ACE inhibitory activities [39,40]; in addition, proline and positively charged amino acids, lysine and arginine, also significantly increased the potential of ACE inhibitory peptides [41]. Furthermore, the hydrophobic amino acids of the N-terminal region of the peptide would have a certain effect on the binding of the ACE active site [42], and the leucine and proline residues were the most frequently occurring amino acids in the structure of other ACE inhibitory peptides [43]. Therefore, the relatively high inhibitory activity of VLPVLS (248.1 μM) could be due to the presence of the hydrophobic amino acid, valine, at the N-terminus of the peptide, or the frequency of hydrophobic amino acids valine, leucine, and proline in the peptide sequence. In addition, glutamine presented at the C-terminus of the peptides had been reported to be an effective factor in ACE inhibitory activity of ACE inhibitory peptides [40,44]. Glutamine is a neutral form of glutamic acid, thus, glutamine could also chelate with zinc ions at the ACE active site for increasing ACE inhibitory capacity of these peptides herein, including YPQ, AVQ, and AYLQ.

### 3.7. Quantification of Identified Peptides from DDSG-PI Hydrolysate

In the quantitative analysis of peptides, the best MS standard is to use the selective reaction monitoring (SRM) or MRM mode in ESI-QQQ-MS with its highest sensitivity [45]. To quantify the ACE inhibitory peptides derived from DDSG-PI, synthetic peptide standards for external calibrations were used by ESI-QQQ-MS. The optimal MS parameters of synthetic peptide standards were presented in Table 5.

The contents of individual peptides were between 0.1700 and 16.96 mg/g DDSG-PI DW. VLPVLS was the most abundant peptide (16.96 mg/g DDSG-PI DW), followed by YPQ (5.070 mg/g DDSG-PI DW), NQL (5.820 mg/g DDSG-PI DW), AVQ (4.070 mg/g DDSG-PI DW), VLPSLN (1.140 mg/g DDSG-PI DW), and AYLQ (0.1700 mg/g DDSG-PI DW). The presence of VLPVLS, AVQ, and YPQ could have the greatest impact on the potential antihypertensive effects in combination with the concentration of IC_50_ for ACE.

## 4. Conclusions

This study evaluated the ACE inhibitory capacity of ultrafiltration and RP-HPLC fractions derived from DDSG-PI hydrolysates, which were produced by Alcalase, Neutrase, and Flavourzyme. Six novel peptides—VLPVLS, YPQ, NQL, AVQ, VLPSLN, and AYLQ—were successfully identified by UPLC–Q–TOF–MS/MS for the first time from the Alcalase-generated DDSG-PI hydrolysates, with low IC_50_ values for ACE ranging from 181.0 to 1245 μM. The presence of these specific amino acid residues in the sequence could be responsible for the ACE inhibitory activity of these peptides. These peptides in DDSG-PI were quantified by using ESI-QQQ-MS. The results indicated that DDSG could be used as a functional ingredient for further enhancing the value of this underutilized cereal material. In addition, the evaluation of ACE inhibitory activity reveal the potential possibility of these peptides to treat hypertension. However, whether these identified peptides have antihypertensive effects in vivo remains to be investigated in the next study.

## Figures and Tables

**Figure 1 foods-08-00390-f001:**
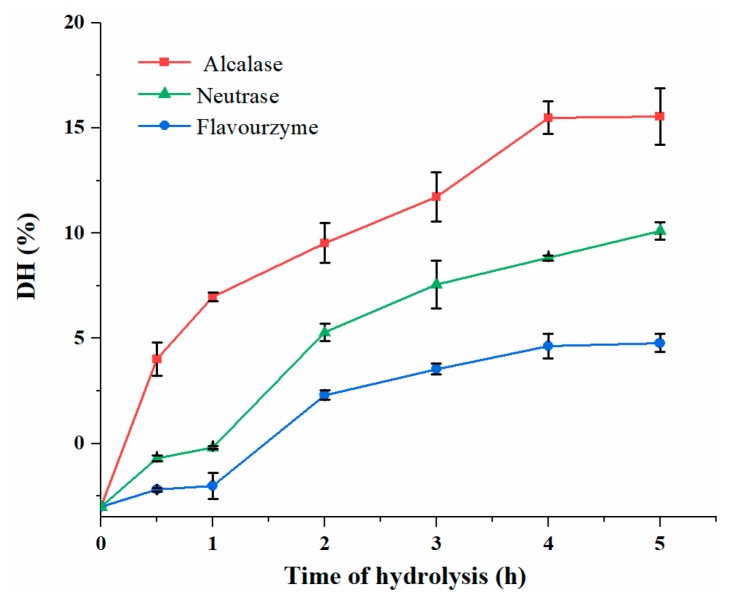
Degree of hydrolysis (DH) of DDSG-PI hydrolysates treated by three proteolytic enzymes.

**Figure 2 foods-08-00390-f002:**
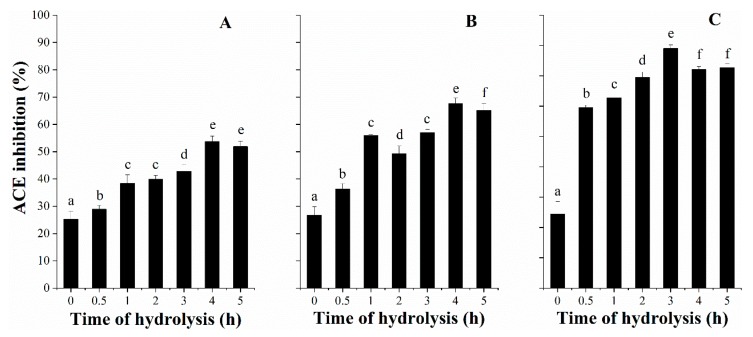
Angiotensin converting enzyme (ACE) inhibitory activity of DDSG-PI hydrolysates as a function of hydrolysis time using different proteases. Key: (**A**), Flavourzyme; (**B**), Neutrase; (**C**), Alcalase. Hydrolysates were taken at 0, 0.5, 1, 2, 3, 4, and 5 h, freeze-dried, and then ACE inhibitory activity was measured at a final concentration of 1 mg/mL (based on dry weight (DW)). All data were presented as means ± SD (*n* = 3). Samples with different letters in the same sample group showed significant differences *(p* < 0.05).

**Figure 3 foods-08-00390-f003:**
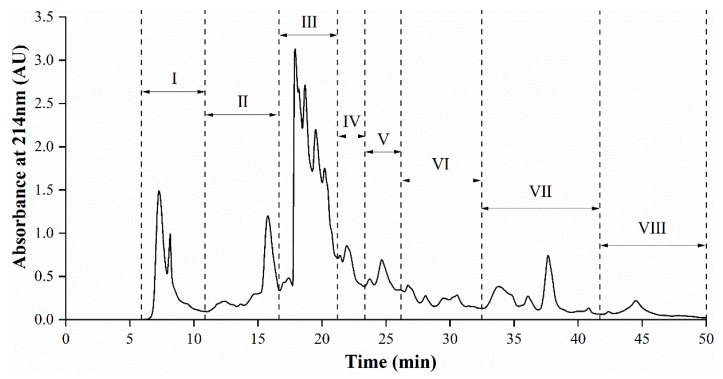
Reversed phase high performance liquid chromatography (RP-HPLC) elution profile of Alcalase-generated DDSG-PI hydrolysate (<1 kDa) after ultrafiltration. The eluates were collected by every tube and combined into eight different fractions labeled I–VIII.

**Figure 4 foods-08-00390-f004:**
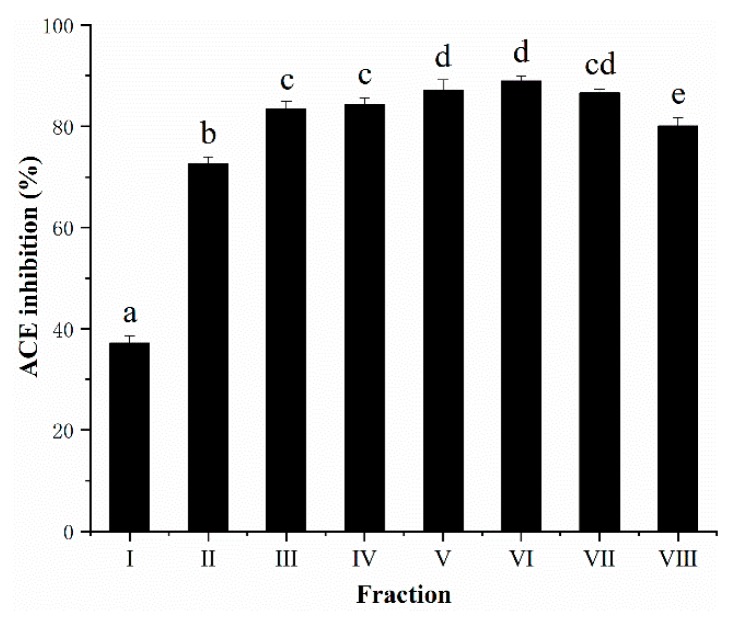
ACE inhibition of different fractions after semi-preparative RP-HPLC. Elutes were combined into eight fractions and analyzed for ACE inhibitory activity at a final concentration of 0.5 mg/mL (based on DW). All data were presented as means ± SD (*n* = 3). Different letters indicated the significant difference at *p* < 0.05.

**Figure 5 foods-08-00390-f005:**
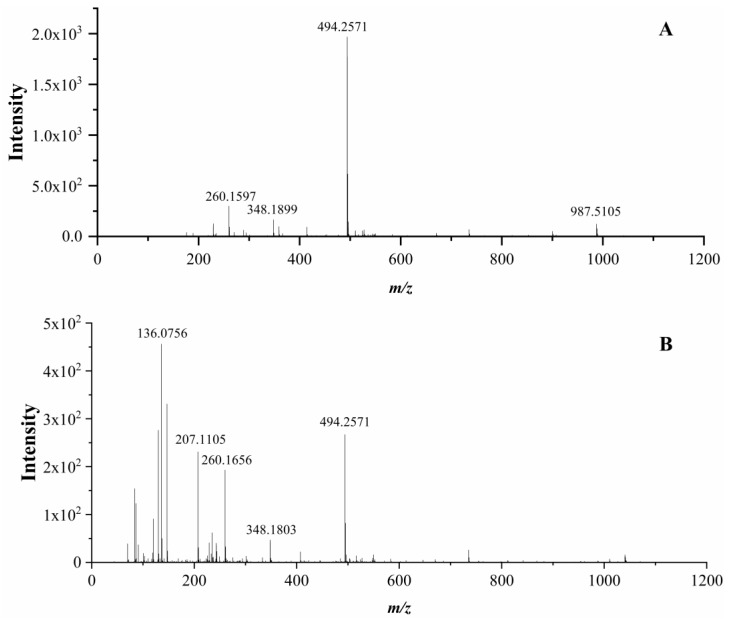
Mass spectrometry (MS) spectra of peak at 14.08 min by full scan-survey mode. Key: (**A**), at low collision energy (6 eV); (**B**), at high collision energy (20 eV).

**Figure 6 foods-08-00390-f006:**
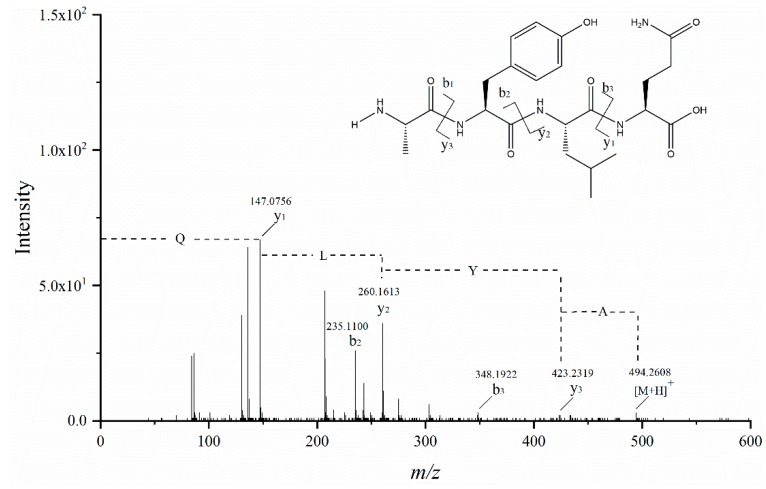
MS/MS spectrum of the peptide (*m/z* = 494.2608) in a single charge mode.

**Table 1 foods-08-00390-t001:** Optimized conditions of quadrupole time-of-flight mass spectrometry (Q–TOF–MS).

Items	MS Mode ^1^	MS/MS Mode ^2^
Capillary voltage	3500 eV	3500 eV
Mass range	20–2000 *m/z*	60–1000 *m/z*
Acquisition rate	4 spectra/s	2 spectra/s
Cone voltage	20 eV	20 eV
Source temp.	100 °C	100 °C
Desolvation temp.	400 °C	400 °C
Desolvation gas flow	700 L/h	700 L/h
Cone gas flow	50 L/h	50 L/h
Collision energy ^3^	6/20 eV	10–25 eV

^1^ MS mode: All ions passed through quadrupole and were fragmented in collision cell. ^2^ MS/MS mode: Only target ions passed through the quadrupole. ^3^ In MS mode, the parent ions were found by using different collision energies; In MS/MS mode, the collision energy ranging from 10 to 25 eV was used to generate more fragment ions.

**Table 2 foods-08-00390-t002:** Amino acid composition of dried distilled spent grain prolamin isolates (DDSG-PI)

Amino Acid	Content (mg/g DDSG-PI DW) ^1^
Ala	98.50 ± 2.3
Cys	<0.50
Asp	55.40 ± 0.20
Glu	250.8 ± 22.7
Phe	56.20 ± 0.10
Gly	13.00 ± 0.30
His	10.50 ± 0.30
Ile	40.90 ± 2.7
Lys	<0.50
Leu	154.3 ± 33.2
Met	12.80 ± 0.50
Pro	88.50 ± 11.6
Arg	13.50 ± 0.10
Ser	36.20 ± 5.6
Thr	18.80 ± 0.80
Val	47.20 ± 6.4
Tyr	37.50 ± 0.70
Total	940.0

^1^ All data were presented as means ± SD (*n* = 3).

**Table 3 foods-08-00390-t003:** Concentrations inducing 50% inhibition (IC_50_) values for angiotensin converting enzyme, and the yield from Alcalase-generated hydrolysate ultrafiltration from DDSG-PI ^1^.

Fractions	IC_50_ (μg/mL)	Yield (%)
AGH	11.92 ± 0.18	100.0
>5 kDa	24.58 ± 1.43 *	3.920
3–5 kDa	10.41 ± 0.79 *	2.550
1–3 kDa	26.74 ± 1.02 *	9.350
<1 kDa	6.750 ± 0.37 *	84.18

All data were presented as means ± SD (*n* = 3). * presented the significant difference from the Alcalase-generated hydrolysate (AGH) at *p* < 0.05.

**Table 4 foods-08-00390-t004:** Information of identified peptides and the concentration inducing 50% inhibition (IC_50_) for ACE of corresponding synthetic peptides ^1^.

Peptide Sequence	No. of Amino Acid Residues	Theoretical Mass (Da)	Observed Molecular Ion, *m/z* (Charge)	IC_50_ (μM)
AVQ	3	316.1747	317.1808	181.0 ± 6.17 ^a^
NQL	3	373.1961	374.2036	704.6 ± 16.06 ^b^
YPQ	3	406.1852	407.1905	220.0 ± 9.56 ^c^
AYLQ	3	493.2536	494.2571	228.6 ± 9.63 ^c^
VLPVLS	3	626.4003	627.4010	248.1 ± 7.15 ^d^
VLPSLN	3	641.3748	642.3737	1246 ± 9.92 ^e^

^1^ All data were presented as means ± SD (*n* = 3). Different letters in the same sample group indicated the significant difference at *p* < 0.05.

**Table 5 foods-08-00390-t005:** Optimal MS parameters of peptide ion pairs using Intellistart in multiple reaction monitoring (MRM) mode and amounts of ACE inhibitory peptides (mg/g) derived from DDSG-PI ^1^.

Compound	Precursor ion (*m/z*)	Cone Voltage (V)	Product ion (*m/z*)	Collision Energy (V)	Content (mg/g DDSG-PI DW)
AVQ	317.1	16	147.2	12	4.070 ± 0.12
NQL	374.3	18	242.8	14	5.820 ± 0.85
YPQ	407.1	26	244.1	20	5.070 ± 0.18
AYLQ	494.5	26	135.5	40	0.1700 ± 0.02
VLPVLS	627.4	28	169.2	40	16.96 ± 1.71
VLPSLN	642.4	28	430.3	22	1.140 ± 0.12

^1^ The quantitative data were presented as means ± SD (*n* = 3).

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
