# Peer review of "In Vitro Production and Identification of Angiotensin Converting Enzyme (ACE) Inhibitory Peptides Derived from Distilled Spent Grain Prolamin Isolate"

_foods, 2019, doi:10.3390/foods8090390_

Round 1

Reviewer 1 Report

This research article is well written and provides interesting and informative scientific findings from distilled spent grain (DSG), the biggest by-product of the Chinese liquor industry. It shows the potential of prolamines from DSG in production of ACE-inhibitory peptides. However there are few doubts that need to be corrected for better comprehension to the readership.

Comments & revision:

Section 2.1: Could the authors provide more specific information regarding the composition of DSG? For example, the % of rice or wheat or barley in the DSG.

This is very crucial because it could affect the judgement of the protein identification and the integrity of raw materials.

Section 2.3: Protein content was measured by macro-Kjeldahl and the protein conversion factor used was 6.25. This doesn’t make any sense since the raw material used was likely the origin of cereals. The protein conversion factor should be around 5.8. Please double check it. Section 2.13: “The statistical analysis and comparison was……” should be “were” Section 3.1: Could the authors provide the information on the protein content of DDSG and the yields of DDSG and DDSG-PI? These are the key information when it comes to commercial production. Additionally, the total amino acids content doesn’t make sense. If it is 940 mg/g, the total amino acids content accounts for 94% whereas the protein content of prolamines was only 4.9% (49.06 mg/g). I strongly suggest the authors to check the raw experimental data and my guess is that the unit used should be “mg/100g” for amino acids composition.

Author Response

Response to Reviewer 1 Comments

Point 1 Section 2.1: Could the authors provide more specific information regarding the composition of DSG? For example, the % of rice or wheat or barley in the DSG. This is very crucial because it could affect the judgement of the protein identification and the integrity of raw materials.

Response 1:The information about the composition of DSG was added in line 48-51. 

Point 2  Section 2.3: Protein content was measured by macro-Kjeldahl and the protein conversion factor used was 6.25. This doesn’t make any sense since the raw material used was likely the origin of cereals. The protein conversion factor should be around 5.8. Please double check it.

Response 2: Protein conversion factor 5.8 was corrected in line 96 and responding result was corrected in line 223.

Point 3 Section 2.13: “The statistical analysis and comparison was……” should be “were”

Response 3: changed.

Point 4 Section 3.1: Could the authors provide the information on the protein content of DDSG and the yields of DDSG and DDSG-PI? These are the key information when it comes to commercial production. Additionally, the total amino acids content doesn’t make sense. If it is 940 mg/g, the total amino acids content accounts for 94% whereas the protein content of prolamines was only 4.9% (49.06 mg/g). I strongly suggest the authors to check the raw experimental data and my guess is that the unit used should be “mg/100g” for amino acids composition.

Response 4: The study of the basic ingredients of DDSG will be another part of our research; we have rechecked the results carefully. The purpose of protein content analysis was to know the prolamines content in DDSG and the purity of extracted prolamines, and the purpose of amino acid composition analysis was to understand the distribution of hydrophobic amino acids in pure prolamines extracted from DDSG to further evaluate the potential of generating ACE inhibitory peptides.

Reviewer 2 Report

1. Please explain the reasons for using the exact enzymes mentioned in the study; the enzyme EC numbers (when known) should be added.

2. Please add a reference number to Conolly et al (line 276, page 8)

3. Please complete the methods with the description of the tool involved to calculate theoretical masses of peptides;

4. BIOPEP datatabse mentioned in the study is called BIOPEP-UWM database. Please modify it. The reference papers for AHPDB and BIOPEP-UWM databases are missing. Please also add the accession dates when using these databases.

What were the search options applied when using these databases in order to find peptides of interest? Are there any data about additional bioactivity of these peptides?

If you applied a nagative selection (meaning: no peptide in database = we discovered a novel one), is it possible they can be published in other databases? What is your opinion about this issue?

5. According to nomenclature and literature, precursor ions are defined using capital letters (e.g Y ion, not an  y ion). Please consider this when modyfing this part of your manuscript.

6. You indicated which spent grain hydrolysate had higher ACE inhibition. Was it reflected also by the presence of peptides in it? Were these peptides found in all hydrolysates?

Author Response

Point 1: Please explain the reasons for using the exact enzymes mentioned in the study; the enzyme EC numbers (when known) should be added.

Response 1: Based on the literature reference (e.g. ref 12 and 31 in this paper) and the results of previous pre-experiment for enzymatic hydrolysis followed by the assay for ACE inhibitory activity, we found that these exact enzymes have the best hydrolysis effect and the high ACE activity of the corresponding hydrolysates. The enzyme EC numbers were added in line 71-73.

Point 2: Please add a reference number to Connolly et al (line 276, page 8).

Response 2: The reference number was added in line 282.

Point 3: Please complete the methods with the description of the tool involved to calculate theoretical masses of peptides.

Response 3: The description of the Biolynx tool was added in line 187-192.

Point 4: BIOPEP database mentioned in the study is called BIOPEP-UWM database. Please modify it. The reference papers for AHPDB and BIOPEP-UWM databases are missing. Please also add the accession dates when using these databases. What were the search options applied when using these databases in order to find peptides of interest? Are there any data about additional bioactivity of these peptides? If you applied a negative selection (meaning: no peptide in database = we discovered a novel one), is it possible they can be published in other databases? What is your opinion about this issue?

Response 4: The name of database was corrected in line 345-347, and the references were added in line 345-347. The accession dates were added in line 345-347. Currently, these two databases contain the most comprehensive information of bioactive peptides and antihypertensive peptides. According to the search results, there were no reports on other additional bioactivity of these peptides, and this will be next research plan. We searched these two peptide databases combined with others including Web of Science, Google scholar, Scifinder, Chemspider, et al. and no reports of related peptides were found, so we believed these identified peptides were novel.

Point 5: According to nomenclature and literature, precursor ions are defined using capital letters (e.g Y ion, not an y ion). Please consider this when modifying this part of your manuscript.

Response 5: We have carefully rechecked our paper, but b and y ions appeared in our paper were product ions, not precursor ions.

Point 6: You indicated which spent grain hydrolysate had higher ACE inhibition. Was it reflected also by the presence of peptides in it? Were these peptides found in all hydrolysates?

Response 6: Because the idea of the experiment was based on activity-orientation, relatively speaking, the higher ACE inhibition means the presence of peptides with higher ACE inhibitory activity. There might be other peptides present in the hydrolysates, but we did not find peptides with high ACE inhibitory activity according to our pre-experiment.